# Inhibition of endogenous ouabain by atrial natriuretic peptide is a guanylyl cyclase independent effect

**Gulay Tegin[1], Yonglin Gao[1], John M. Hamlyn[2], Barbara J. Clark[3], Rif S. El-Mallakh[1]***

**1** Department of Psychiatry and Behavioral Sciences, University of Louisville, Louisville, Kentucky, United States of America, **2** Department of Physiology, School of Medicine, University of Maryland, Baltimore, Mississippi, United States of America, **3** Department of Biochemistry and Molecular Genetics, University of Louisville, Louisville, Kentucky, United States of America

* rifaat.elmallakh@louisville.edu

**Data Availability Statement:** Data are available on Zenodu under the title and authors of this publication, and may be accessed at: https://

## Abstract

### Background

Endogenous ouabain (EO) and atrial natriuretic peptide (ANP) are important in regulation of sodium and fluid balance. There is indirect evidence that ANP may be involved in the regulation of endogenous cardenolides.

### Methods

H295R are human adrenocortical cells known to release EO. Cells were treated with ANP at physiologic concentrations or vehicle (0.1% DMSO), with or without guanylyl cyclase inhibitor 1,2,4 oxadiazolo[4,3-a]quinoxalin-1-one (ODQ). Cyclic guanosine monophosphate (cGMP), the intracellular second messenger of ANP, was measured by a chemiluminescent immunoassay and EO was measured by radioimmunoassay of C18 extracted samples.

### Results

EO secretion is inhibited by ANP treatment, with the most prolonged inhibition (90 min vs $\leq$ 60 min) occurring at physiologic ANP concentrations (50 pg/mL). Inhibition of guanylyl cyclase with ODQ, also reduces EO secretion. The inhibitory effects on EO release in response to cotreatment with ANP and ODQ appeared to be additive.

### Conclusions

ANP inhibits basal EO secretion, and it is unlikely that this is mediated through ANP-A or ANP-B receptors (the most common natriuretic peptide receptors) or their cGMP second messenger; the underlying mechanisms involved are not revealed in the current studies. The role of ANP in the control of EO synthesis and secretion *in vivo* requires further investigation.

zenodo.org/record/5594568#.YXRY-S2z3a8 (doi: 10.5281/zenodo.5594568).

**Funding:** The authors received no specific funding for this work.

**Competing interests:** I have read the journal's policy and Dr. El-Mallakh is on the speakers' bureaus of Eisai, Indivior, Intra-Cellular Therapies, Janssen, Lundbeck, Noven, Otsuka, Sunovion, and Teva. The other authors of this manuscript do not have any competing interest.

## Introduction

Endogenous cardenolides (ECs) are groups of steroid back-boned chemicals that include ouabain, digoxin, bufodienolides, and their metabolites [1–5]. These chemicals were originally identified from plants and non-mammalian sources including the foxglove (*Digitalis* species) [6], climbing oleander *(Strophantus gratus* and other Apocynaceae) [7], and in toads (*Bufo*) and frogs (*Atelopus*) [8]. More recently, it has become clear that these compounds or very similar mammalian counterparts are synthesized and released from adrenal gland and hypothalamus [2, 9–12]. In the human body, they are thought to play important roles in sodium and fluid homeostasis by affecting sodium- and potassium-activated adenosine triphosphatase ($Na^+$-$K^+$-ATPase) activity, and in cellular signaling pathways [13–17]. Their role in blood pressure control has been more extensively examined.

Endogenous ouabain (EO) has been implicated in many cellular mechanisms and physiologic pathologies [17], including modulation of cellular or organ size [18, 19], sodium homeostasis [20], hypertension [21, 22], mood disorders [23, 24], and possibly cancer [25]. This is accomplished via regulation of sodium pump activity [13–17] and cellular signaling pathways [26–28].

The natriuretic peptide family mainly consists of Atrial Natriuretic Peptide (ANP), Brain or B-type Natriuretic Peptide (BNP), and C-type Natriuretic Peptide (CNP). Each natriuretic peptide in this family appears to induce diuresis, natriuresis, vasodilation, and inhibition of the renin-angiotensin-aldosterone system and the sympathetic nervous system [29–31]. Recent research suggests multiple interactions between cardiotonic steroids (CTS) and ANP [32]. Both ANP and ECs are thought to play important roles in sodium and fluid homeostasis. In cardiac tissue, both endogenous cardenolides and ANP were increased in similar pathological conditions [32]. For example, in a study to test possible interactions between ANP and ouabain in the heart it was found that ANP antagonizes stimulation of the $Na^+$-$K^+$-ATPase by low concentrations of ouabain [33]. However, ANP when used alone stimulated $Na^+$-$K^+$-ATPase activity [33].

There are three known natriuretic peptide receptors: Natriuretic Peptide Receptor-A (NPR-A), NPR-B, and NPR-C. ANP and BNP preferentially bind to NPR-A [34], while NPR-C has 50-fold greater affinity to CNP than either ANP or BNP [35]. Both NPR-A and NPR-B are coupled with guanylyl cyclase [34]. In contrast, NPR-C has no guanylyl cyclase activity [36] and has long been thought to act as a natriuretic peptide clearance receptor because it is primarily found in the kidney, has similar affinity for two of the three natriuretic peptides [35, 37], and its elimination in transgenic mice increases the half-life of ANP by 66% [38]. NPR-C appears to be coupled with the $G_{i\alpha}$ protein and other intracellular signals, but not cGMP [39]. Activation of this receptor with a ring deleted ANP analogue, cANF[4-23], which blocks natural ANP and does not increase cGMP, results in natriuresis and reduction of blood pressure in conscious rats [40]. The NPR-C is widely distributed [36] including in the adrenal gland, where it is 5-fold less abundant than NPR-A [41].

The vasodilatory effects of ANP appear to be mediated by stimulation of guanylyl cyclase, elevation of intracellular cyclic guanosine monophosphate (cGMP), and activation of cGMP-dependent protein kinase [42–44]. ANP increases cGMP production and attenuates agonist-stimulated aldosterone synthesis in adrenal zona glomerulosa cells [42]. Over 30 years ago, ANP was demonstrated to reduce synthesis of a $Na^+$-$K^+$-ATPase inhibitor by brain tissue both *in vivo* and *in vitro* [45].

In the present study, we investigated whether ANP is involved in the secretion of EO in H295R human adrenocortical cells, which have previously demonstrated to spontaneously excrete endogenous cardenolides [12] and which have functional ANP receptors [46]. We

examined the roles of guanylyl cyclase (using 1H-[1,2,4]oxadiazolo[4,3-a] quinoxaline-1-one {ODQ} to inhibit the enzyme) and cGMP in mediating an inhibitory effect of ANP on EO synthesis.

## Methods

### Cell culture

Human adrenocortical cells (H295R) were from American Type Culture Collection (ATCC). NCI-H295R was adapted from the NCI-H295 pluripotent adrenocortical carcinoma cell line (ATCC CRL-10296). While the original cells grew in suspension, the adapted cells were selected to grow in a monolayer with a 2 day doubling time.

The cells were grown and maintained in DMEM/F12 medium (GIBCO, Grand Island, NY) containing 25 mL/L NuSerum type I, 10 g/L ITS culture supplements (Corning, NY), and antibiotics in 75 $cm^2$ flasks at 37 ˚C under a humid atmosphere of 5% $CO_2$–95% air.

### Treatments

For the experiments, flasks were seeded with H295R cells and grown to ~80% confluent density at 37˚C in air containing 5% $CO_2$ (~1 x $10^6$ cells). Culture medium was removed and discarded; 3.0 mL of Trypsin-EDTA solution (HyClone™, Cytiva, Marlborough, MA) were added until the cell layer dispersed (usually within 5 min). Then, 6 mL of complete growth medium were added, and the cell suspension was centrifuged at 125 x g for 5 min. The cell pellet was re-suspended in fresh growth medium. The subculture ratio was 1:3 to 1:4. The media was replaced every 3 days.

H295R human adrenocortical cells were treated with: 1) 0.1% DMSO or ANP (5, 50, 200 pg/mL; California Peptide Research, Inc, Salt Lake City, UT) dissolved in 0.1% DMSO for 30 min, 60 min and 90 min; 2) 0.1% DMSO, 50pg/mL ANP, 10 μM of guanylyl cyclase inhibitor 1,2,4 oxadiazolo[4,3-a]quinoxalin-1-one (ODQ; Cruz Biotechnology, Inc Santa Cruz, CA) for 90min; and 3) pretreated with ODQ for 30min followed by 50 pg/mL ANP for 60 min. Normal serum ANP levels are frequently reported in the mid 30s pg/mL with a range up to 200 pg/mL in renal failure [47–49]. The culture media was collected after different time points and different treatment for EO measurements. We only used no treatment controls, and did not examine the effect of adding other non-ANP peptides to the cell cultures. All experiments had 3–6 replicates.

### Measurements

H295R human adrenocortical cells were treated with ANP at various concentrations (0, 5, 50, 200 pg/mL) for 30, 60 and 90 min in serum-free medium. The culture media were collected for assay of endogenous ouabain (EO) by radioimmunoassay of C18 extracted samples [50]. In brief, 200 mg Bond Elut C18 columns (Varian) were preconditioned by sequential passage of 3 mls each of 100, 50, 25, and 5% $CH_3CN$ in water. Thereafter, each column was washed twice with 3 ml water. For the samples, 10–12 ml of the conditioned cell culture fluid was centrifuged at 2,500 x g for 20 min. The supernatants were applied to the columns which were then washed sequentially with 2 x 3 ml water and 3 ml 5% $CH_3CN$. Differential elution was then used to elute the desired bound steroids as follows: Bound EO was eluted by a 20% $CH_3CN$ wash, while less polar steroids including endogenous digoxin-like immunocrossreactive materials [9, 51] and aldosterone, that remained bound after the 20% $CH_3CN$ wash were eluted with a 50% $CH_3CN$ wash. Sample blanks were run in parallel. In some experiments, the recovery of EO as well as the success of the differential elution was assessed by inclusion of tracer amounts of $^3$H-

ouabain and $^3$H-digoxin. For immunoassay, all eluates of interest were dried by vacuum centrifugation (Savant) and reconstituted at 60–120 x their original concentration in water. For analysis of the efficiency of steroid recoveries, aliquots of the eluates were taken directly for liquid scintillation counting. The radioimmunoassay was performed in a manner similar to that previously described using the R8 ouabain-antiserum [50] with the exception that a micro format was employed wherein the total assay volume was 50 μL. In this assay, 30 μL was the reconstituted sample and the remainder was comprised of $^3$H-ouabain (final concentration = 1.35 nM) and buffer. The reaction was incubated at room temperature for one hour and terminated by rapid filtration over glass fiber filters (Whatman GF/B) using a 24 well cell harvester (Brandel). The filters were soaked in scintillation cocktail (Ultima Gold, Perkin Elmer) for 40 hours and then counted (Beckman TA3000) with quench correction. Non-specific binding was determined by inclusion of excess ouabain in the reaction and typically was < 4% of the total bound count. The antiserum exhibits no meaningful crossreactivity (<0.01%) with the vast majority of common adrenocortical, ovarian, or testicular steroids (see Table 1 in [51]). In addition, the immuno-crossreactivities for dihydro-ouabain, digoxin, digitoxin, and aldosterone were 0.16, 5.2, 28 and 0.012%, respectively. The crossreactivity of the latter three steroids is of no concern; the use of a differential elution process excludes the vast majority of digoxin, digoxin-like materials, and aldosterone from the eluates used for EO measurement. The low crossreactivity for dihydro-ouabain, i.e., 625-fold less than for ouabain, makes it unlikely that any dihydro-ouabain that may have been secreted [11, 12] contributed significantly to the EO measurement. All measurements were made with an operator blinded to the study design.

We did not assay intracellular EO concentrations because previous work revealed that the intracellular EO concentrations in H295R cells was below assay detection [12]. Four mL PBS were added to the remaining cells that were scratched twice, and then transferred to a centrifuge and spun at 125x g for 5 minutes. The cell pellet lysates of were used for intracellular cGMP assay using a chemiluminescent immunoassay kit (Arbor assay, Ann Arbor, Michigan) based on the product manual. Protein was determined by the BCA method (Pierce, Grand Island, NY). Samples were stored at -80˚C.

### Statistics

Student's T-test was used to examine differences in response. The data were presented as mean ± SE.

### Results

Fig 1 shows the integrity of the C18 extraction with respect to ouabain and digoxin. Essentially all of the $^3$H-ouabain added to samples appeared in the 20% eluate with a yield approaching 100%. The small amount of label detected in the flow through (i.e., unbound label) for both ouabain and digoxin likely represents free $^3$H in water. In contrast, the vast majority of the $^3$H-digoxin was detected in the 50% eluate, there being minimal cross contamination (~ 6%) in the 20% eluates.

Fig 2 shows the standard curve for the ouabain radioimmunoassay. The data were fitted by a standard sigmoidal four parameter Hill equation of the form: $f = y0 + a^* x^b / (c^b + x^b)$. The slope of the relationship was -1.069 consistent with a single class of specific binding sites with minimal apparent negative or positive cooperativity under the conditions used. The $EC_{50}$ was 4.83 +/- 0.27 nM. The threshold sensitivity of the assay, defined as 5% displacement of the label from the control value, ranged from 2–5 femtomoles. The coefficient of error for the EO measurement expressed as a % variance of the mean value from three replicates for each sample

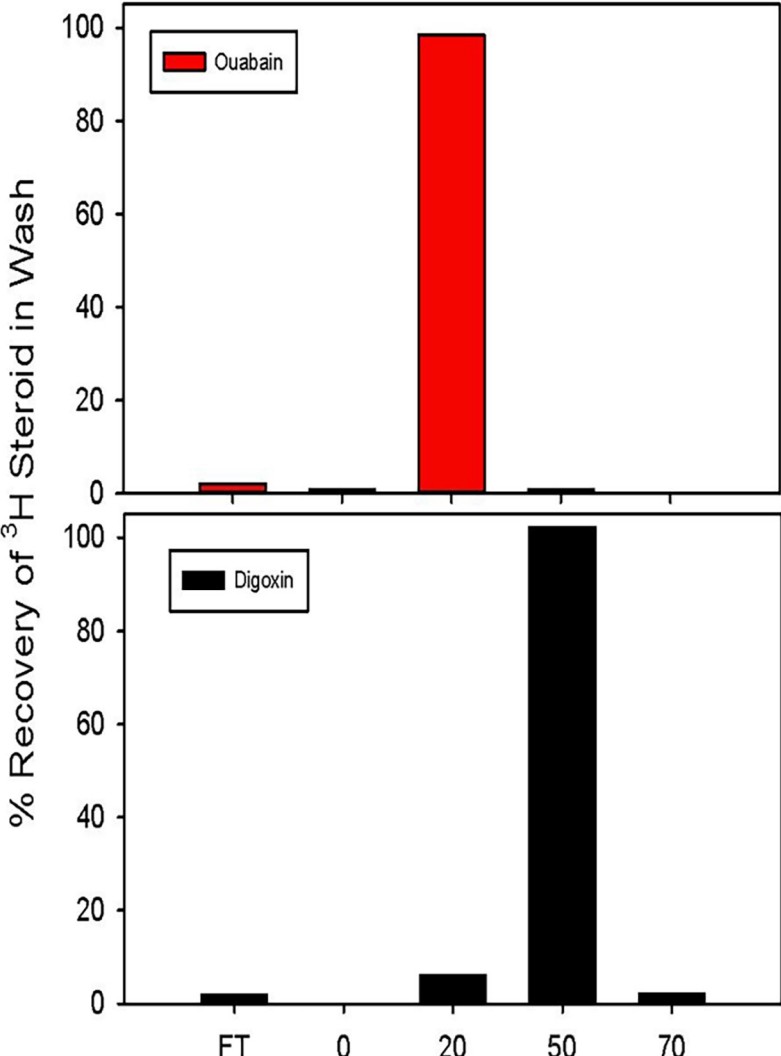

**Fig 1. Elusion of ouabain and digoxin.** A: Extraction and differential elution of $^3$H-ouabain 9upper panel) and $^3$H-digoxin (lower panel) from C18 columns. Labelled steroids were introduced into conditioned cell culture medium and applied to preconditioned C18 columns as described. The flow through (FT) as well as the eluates from the 0, 20, 50 and 70% CH$_3$CN washes were collected and counted. Results are shown as the average of duplicate experiments.

ranged between 0.42 and 5.2%. The assay precision ranged from 94–99% in the working range of the samples.

Human H295R adrenocortical cells grown in culture spontaneously secrete EO at low rates, especially when compared to aldosterone. The concentration in the supernatant spontaneously increases over time, and nearly doubles in 90 minutes (Fig 3). Addition of ANP inhibited the appearance of EO at 60 minutes and 90 minutes of treatment compared to untreated cells (Fig 3). At 60 minutes, all concentrations of ANP tested inhibited EO secretion with no obvious dose-effect (Fig 3). However, at 90 minutes, the inhibitory effect of ANP persisted at the physiological concentration of 50 pg/mL. In contrast, the significant effects of subphysiologic ANP concentrations (5 pg/mL) or supraphysiologic ANP concentrations (200 pg/mL) that were apparent at 60 minutes had waned by 90 minutes ($0.014 \pm 0.003$ vs $0.062 \pm 0.006$, t = 6.58, $P = 0.03$; $0.014 \pm 0.003$ vs $0.053 \pm 0.003$, t = 8.70, $P < 0.001$, respectively) (Fig 3).

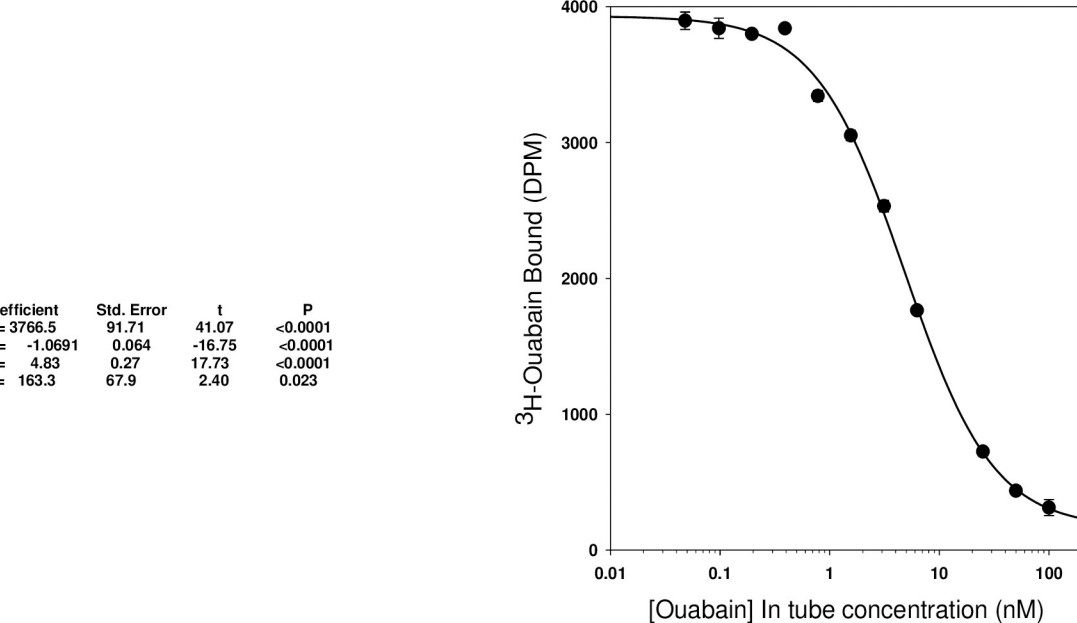

| Coefficient | Std. Error | t | P |
|---|---|---|---|
| a = 3766.5 | 91.71 | 41.07 | <0.0001 |
| b = -1.0691 | 0.064 | -16.75 | <0.0001 |
| c = 4.83 | 0.27 | 17.73 | <0.0001 |
| y = 163.3 | 67.9 | 2.40 | 0.023 |

**Fig 2. ouabain standard curve.** Radioimmunoassay standard curve for ouabain. Each point is the mean ± SEM for triplicate determinations. Error bars are obscured where the symbol diameter > sem. The data were fitted iteratively to the equation described in the text and the resultant parameters including coefficients (a = $B_{max}$, b = slope, c = $EC_{50}$, d = nonspecific binding), standard errors, t statistics, and probability values for the fit are shown below the figure.

The ANP-A receptor mediates its effect through activation of guanylyl cyclase that generates cGMP. However, the generation of cGMP by ANP in H295R cells was muted and dose-dependent. Specifically, cyclic GMP levels increased significantly when the cells were treated with ANP at 50 pg/mL for 60 min (0.084 ± 0.008 vs 0.050 ± 0.011, t = -2.65, $P < 0.05$), but this effect was transient, returning to baseline after 90 min (Fig 4). Based on this result, ANP 50 pg/mL was selected for the following experiments that explored the role of guanylyl cyclase.

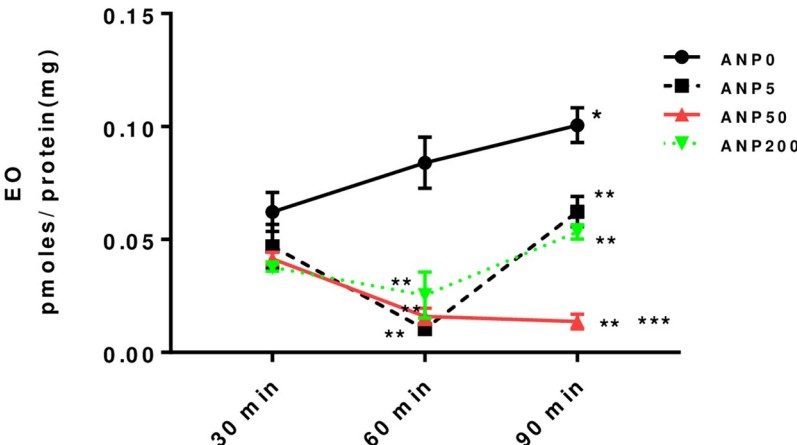

**Fig 3. EO content in culture media.** The EO content in the culture media of H295R human adrenocortical cells increased significantly at 90 min. in the absence of ANP treatment. ANP concentrations of 5, 50, 200 pg/mL reduced EO secretion at 60 and 90 min. The inhibitory effect of 50 pg/mL ANP was significantly lower compared to lower (ANP5) and higher (ANP200) at 90 min. Error bars represent standard error. (*$P < 0.05$, compared to 30 min untreated, **$P < 0.05$, compared to untreated at corresponding time point, ***$P < 0.05$, compared to ANP5 and ANP200 at 90 min. Y axis units are pM/mg protein).

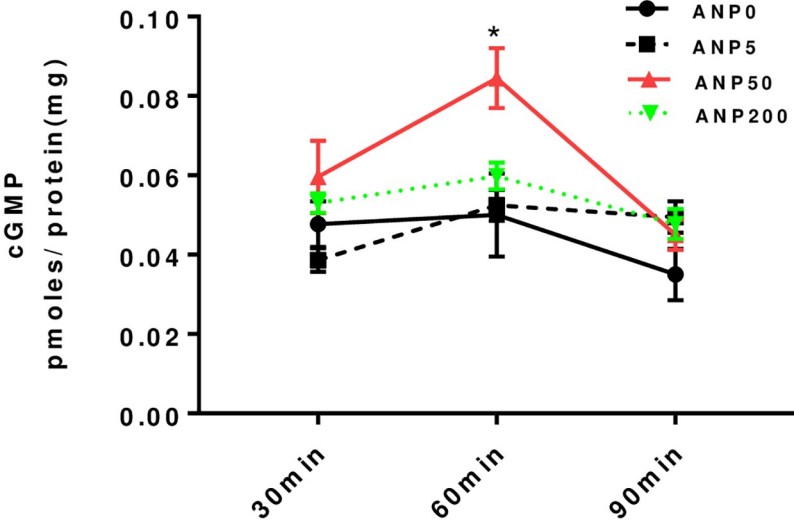

**Fig 4. cGMP Levels.** Intracellular cGMP levels following ANP treatment for 30, 60, and 90 min in H295R human adrenocortical cells. A significant increase in cGMP was observed only with 50 pg/mL at 60min. Error bars represent standard errors. ($^*P < 0.05$ compared to control; Y axis units are pM/mg protein).

EO secretion by human H295R adrenocortical cells was significantly reduced after treatment with ANP 50 pg/mL for 60 min (untreated 0.145 ± 0.003 vs ANP 50 pg/mL 0.067 ± 0.004 pM/mg protein; $P < 0.05$) and to a similar degree when 10 μM of the guanylyl cyclase inhibitor, ODQ, was used alone (untreated 0.145 ± 0.003 vs ODQ 10 μM 0.092±0.012 pM/mg protein; $P < 0.05$) (Fig 5). The combination of the two treatments, pretreatment with ODQ 10 μM for 30 minutes followed by ANP 50 pg/mL for 60 minutes, suppressed EO secretion further when compared with untreated or ANP alone or ODQ alone (to 0.03 ± 0.006 pM/mg protein, $P < 0.05$ for both) (Fig 5).

While we use the term 'secretion', our data does not determine the mechanism involved or the intracellular source of the secreted EO that was measured. The present experiments do not

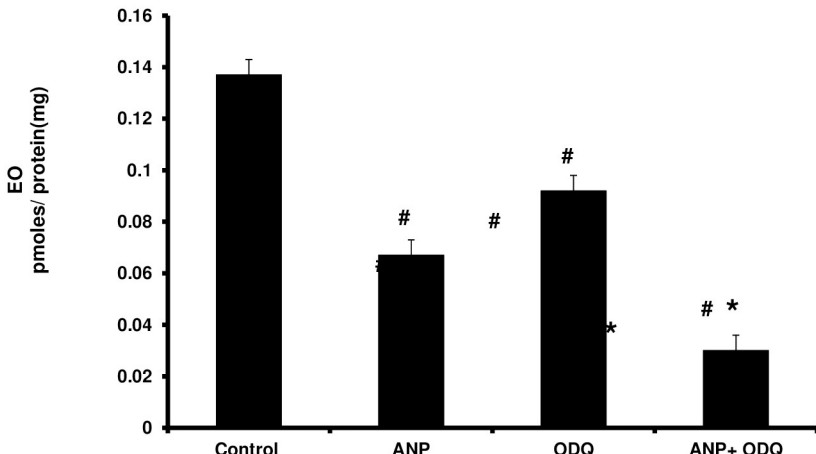

**Fig 5. Effect of ANP on EO secretion.** Change in EO secretion 60 min following addition of ANP 50 pg/mL, 10 mM ODQ and ODQ+ANP treatment. ANP alone and ODQ alone reduced EO secretion ($^\#P < 0.05$ compared to control). Pretreatment with ODQ followed by ANP further decreased secreted EO when compared to ANP or ODQ alone ($^*P < 0.05$). (Error bars represent standard error; Y axis units are pM/mg protein).

determine whether the spontaneous secretion of EO from the unstimulated cells is derived from de novo synthesis, as seems likely, or is the result of the release of stored EO, or even if there is a degradation of a carrier protein that obscures EO measurements at baseline. Likewise, the ability of ANP to suppress the basal EO secretion may involve one or more of the phenomena mentioned above.

Data has been submitted to Zenodo under the title and authors of this publication (and may be accessed at: https://zenodo.org/record/5594568#.YXRY-S2z3a8).

## Discussion

EO is produced in the adrenal and to some limited extent in the hypothalamus [2, 9, 11, 12]. Human H295R adrenocortical cells also produce and secrete small amounts of EO in culture [12]. In the present work, we have replicated earlier observations concerning the spontaneous production of an ouabain-like compound by these cells in culture (Fig 3). We have also demonstrated that ANP at physiologic concentrations inhibits the production of EO (Fig 3). However, our observations are of some surprise because they indicate that the inhibitory effect of ANP on EO secretion is unlikely to be mediated through the ANP-A receptor, which is the most common natriuretic peptide receptor [41], and utilizes cGMP as a second messenger [34]. In our studies, inhibition of generation of cGMP did not prevent the ANP effect, and actually reduced EO production independent of ANP (Fig 5). Further, the effects of ANP and of guanylyl cyclase inhibition appear to be additive (Fig 5), suggesting that they are mediated by independent mechanisms. Nonetheless, there are potential alternative explanations of the data. For example, nitric oxide (NO) is known to activate guanylyl cyclase and sufficient NO can overcome ODQ inhibition [52]. Likewise, confirmation of a guanylyl cyclase component does not exclude other potential mechanisms of ANP, as is the case with NO [44, 53]. This is especially important given the lack of a clear dose response effect of ANP on either EO or cGMP production.

Given that inhibition of guanylyl cyclase does not reverse the EO inhibition caused by ANP and may actually magnify it (Fig 5), the control of EO synthesis may be mediated in part by NPR-C. This conclusion is further supported by the minimal and transient increase of cGMP seen in the cells (Fig 4). Conversely, the transient nature of effects of ANP on both EO and cGMP may be consistent with the apparent tachyphylaxis seen in blood pressure control with ANP [54].

The existence of endogenous cardenolides, like EO, remains controversial to some [55] despite overwhelming evidence of its existence and physiological and pathological roles—some of which has been proven in studies using transgenic animals [21]. Many of the challenges related to the successful measurement of EO have been described elsewhere [21]. Of critical importance is the extraction procedure which must be performed with care. Ouabain is a highly polar steroid; its interaction with C18 is much weaker than all of the known adrenocortical steroids and most of the known CTS. Accordingly, it is critical to fully precondition the C18 columns, bringing them carefully and fully to water before sample application to ensure that all of the sample EO will become bound. The presence of even small amounts of organic solvents such as acetonitrile or methanol will prevent EO from binding and it will be lost to the flow through. Further, the mass of the C18 sorbent must be sufficient to trap all the EO and this is a function of the sample mass, type, and origin. For example, 200 mg of C18 sorbent will typically trap all the EO from 2–3 ml of plasma. With conditioned cell culture medium, up to 20 ml can be extracted per column because this matrix is much less complex than native plasma and the competition of ancillary polar chemical entities with EO for binding to the sorbent is much less. Once the sample has been applied and EO is bound, it is

necessary to wash with the appropriate volumes of water and then low concentrations of aceto-nitrile (2.5 to 5%) or methanol (5–10%) to reduce some of the very highly polar entities that can interfere in the radioimmunoassay and lead to massive suppression of ionization in mass spectrometry. The volume of this organic wash also is critical; overly large wash volumes will eventually elute bound EO, and it will be lost. Similarly, the concentration and volume of ace-tonitrile used to elute EO is important. Typically, 20–25% acetonitrile is used and the volume is chosen to ensure that all the EO is recovered without significant contamination by e.g., digoxin (Figs 1 and 2) or other less polar eCTS. In validating the extraction method, it is most useful to use $^3$H labeled steroids of interest using the anticipated sample conditions. For the immunoassay, a highly selective antiserum with affinity for ouabain in the high picomolar to low nanomolar range is required that has minimal crossreactivity with the common classical mammalian steroids. Thus far, this antibody requirement has been met with polyclonal but not monoclonal sources. The final critical factor in the radioimmunoassay method we use concerns the method by which the binding assay is terminated. With the vast majority of high affinity ouabain antibodies, the half time for the dissociation of ouabain or EO from antibody binding sites is surprisingly rapid ranging from 2–15 minutes. Thus, the method used to sepa-rate bound from free must be much faster than the dissociation half time if the high sensitivity required to detect EO is to be retained. Among the various options, filtration over glass fiber filters is convenient and very rapid, typically requiring <15 seconds including multiple washes. Another advantage of the method is the very low background binding of the tracer $^3$H-ouabain to the filters. Other assay formats including ELISA [51, 56], radioimmunoassays that do not require a bound/free separation as well as mass spectrometry [21] can be used with the appro-priate modifications.

In the adrenal glands, EO is synthesized in and secreted from the cortex, which also synthe-sizes cortisol, aldosterone, and a number of other steroids. Like aldosterone, EO biosynthesis requires progesterone [57]. However, in the normal adrenal, the amount of EO secreted is some 20–50 times lower than the amount of aldosterone secreted, and some 10,000-fold lower than the amount of cortisol made [21]. Although H295R cells maintain their aldosterone secre-tory phenotype in prolonged cell culture, their ability to secrete EO is reduced ~10–20 fold when compared with unstimulated adrenocortical cells in primary cell culture [56]. Hence, when working with H295R cells and when feasible it is helpful to extract large volumes of cell conditioned media and, critically, to concentrate the samples post extraction prior to immuno-assay. In the present work, the extracted samples were concentrated between 60–120 fold prior to introduction into the assay. In order to obtain interpretable data under these conditions, EO was differentially eluted from the C18 columns to minimize assay cross-contamination with all of the classical adrenocortical steroids. Of particular relevance to the present experi-mental conditions, the differential elution minimizes aldosterone, digoxin-like materials and other non-polar eCTS in the samples used for the EO measurement. In addition, a highly selective ouabain antiserum was used to minimize interference from any dihydro-ouabain that may have been present in the EO extract [11, 12]. Under these conditions, it was possible to explore the impact of ANP and its related signaling on basal EO secretion with reasonable confidence.

A number of studies have previously implicated both angiotensin II and ACTH *in vitro* and more especially ACTH *in vivo* as stimulants to EO secretion [21]. However, investigation of the hormonal factors that might suppress EO secretion are very few. In particular, there are three reports regarding the influence of ANP on eCTS-like activity and these have led to differ-ent conclusions. One study indicated that ANP suppressed the secretion of an eCTS-like factor from brain fragments *in vitro* [45] while two others suggested that central infusions of ANP *in vivo* increased eCTS-like activity in the circulation [58]. Further, the latter response was

blocked by electrolytic lesions in the anteroventral third ventricular region of the brain [59]. It may be noted that all of these studies used bioassays where the nature of the eCTS-like factor was unknown. The new results from the present study, made with extracted samples and a highly selective ouabain-antiserum, confirm that physiological concentrations of ANP do have a modulatory role and significantly suppress EO secretion from adrenocortical cells *in vitro*. But the question remains regarding the significance of this phenomenon. Is the suppressive action of ANP on EO secretion sufficiently dramatic to reduce the circulating levels of EO and lower blood pressure *in vivo*? Various experimental models are available to test this idea including animals transgenic for various ANP receptor subtypes and the rodent model of ACTH induced hypertension. In the latter model plasma EO is elevated and the elevated blood pressure depends upon the ouabain binding site of the sodium pump [60].

The roles of EO and ANP regarding blood pressure and sodium control overlap considerably [20, 34, 41, 60–62] and overlap in their role in pathological states [32], so that one would expect that they may play a role in modulating each other [34]. In this study we found evidence that ANP inhibits the production of EO, and that mechanism does not appear to involve activation of NPR-A or NPR-B receptors. The implication, not tested here due to preconceived expectations about NPR-C, is that NPR-C receptors contribute to the control of EO secretion. Accordingly, future work will be needed to determine the inhibitory mechanism of ANP including the specific role of $G_{i\alpha}$ proteins.

In the present study ANP suppressed the secretion of EO from H295R cells. The effects of ANP we observe could involve an action on the biosynthesis of EO, its secretory mechanism, or both. Our experimental design does not discriminate among these possibilities. However, it is a generally accepted assumption in most all steroid-related studies that secretion occurs by passive diffusion across the plasma membrane and, accordingly, that altered secretion is a simple reflection of corresponding changes in biosynthesis. In the adrenal cortex, it is widely understood that steroids are secreted as they are synthesized and, with the exception of cholesterol esters, are not stored.

As mentioned above, the normal adrenal cortex secretes very little EO when compared with aldosterone. This phenomenon is mirrored also in H295R cells and measurements of EO secretion under basal secretory conditions add further to the challenge. Nevertheless, the suppressive effects of ANP on EO secretion we observed are noteworthy, appear to be independent of intracellular cGMP and are generally consistent with the effects of ANP on the secretion of other adrenocortical steroids. Accordingly, further studies will be needed to further dissect the site of action of ANP on EO secretion under baseline and stimulated conditions and especially when the flow of carbon from cholesterol to aldosterone is blocked.

## Author Contributions

**Conceptualization:** Yonglin Gao, Rif S. El-Mallakh.

**Data curation:** Yonglin Gao.

**Formal analysis:** Yonglin Gao.

**Funding acquisition:** Rif S. El-Mallakh.

**Investigation:** Gulay Tegin, Yonglin Gao, John M. Hamlyn, Barbara J. Clark.

**Methodology:** Yonglin Gao, John M. Hamlyn, Barbara J. Clark, Rif S. El-Mallakh.

**Resources:** Barbara J. Clark.

**Supervision:** Yonglin Gao, Rif S. El-Mallakh.

**Writing – original draft:** Gulay Tegin, Yonglin Gao.

**Writing – review & editing:** John M. Hamlyn, Barbara J. Clark, Rif S. El-Mallakh.

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
