## [Decision Letter · Decision Letter 0]

29 Jun 2021

PONE-D-21-16196

Inhibition of Endogenous Ouabain by Atrial Natriuretic Peptide is a Guanylyl Cyclase Independent Effect

PLOS ONE

Dear Dr. El-Mallakh,

Thank you for submitting your manuscript to PLOS ONE. After careful consideration, we feel that it has merit but does not fully meet PLOS ONE’s publication criteria as it currently stands. Therefore, we invite you to submit a revised version of the manuscript that addresses the points raised during the review process. Several comments on the methodology used and the results were raised by the reviewers.

We look forward to receiving your revised manuscript.

Kind regards,

Luis Eduardo M Quintas, Ph.D.

Academic Editor

PLOS ONE

Journal Requirements:

'I have read the journal's policy and Dr. El-Mallakh is on the speakers' bureaus of Eisai, Indivior, Intra-Cellular Therapies, Janssen, Lundbeck, Noven, Otsuka, Sunovion, and Teva. The other authors of this manuscript do not have any competing interest.'

Additional Editor Comments (if provided):

Reviewers' comments:

Reviewer's Responses to Questions

**Comments to the Author**

1. Is the manuscript technically sound, and do the data support the conclusions?

Reviewer #1: Partly

Reviewer #2: Yes

2. Has the statistical analysis been performed appropriately and rigorously? 

Reviewer #1: Yes

Reviewer #2: Yes

3. Have the authors made all data underlying the findings in their manuscript fully available?

Reviewer #1: Yes

Reviewer #2: Yes

4. Is the manuscript presented in an intelligible fashion and written in standard English?

Reviewer #1: Yes

Reviewer #2: Yes

5. Review Comments to the Author

Reviewer #1: The authors present an interesting work on endogenous ouabain release in vitro.

The main concern for me is the method used to determine the ouabain concentration. The authors refer to it as radioimmunoassay and cite a paper from 1994. Ouabain is a steroid compound and it is difficult to measure by immune detection.

Let me be very clear; it is not that I don't believe the authors. I do. I also want to say that I think the findings which the authors note are probably "real" and worth communicating. That said, I think the methodology for blood processing and ouabain measurement should be very clearly stated. Specifically, this reviewer would like to see....

1. Information on the antibodies used for the measurements

2. Standard curves for ouabain measurement used in assigning values to plasma values.

3. Method for anticoagulation of blood samples and details regarding the handling of samples prior to measurement.

4. Some analysis of elutation fractions from the C18 columns loaded with pooled samples. This should be done with and without spiking with reference ouabain.

5. Greater discussion of the challenges inherent in the immunoassay of ouabain.

The authors describe that the cells were treated with 0.1% DMSO or ANP. Was the ANP given also in DMSO solution? This sentence suggest otherwise.

Reviewer #2: This study by Tegin et al. describes the effect of ANP on the levels of endogenous ouabain in the media of H295R cells. The possible involvement of Guanylyl cyclase in this effect was addressed by studying the effect of ANP following pretreatment with Guanylyl cyclase inhibitor ODQ.

Although the paper describes interesting observations several key issues have to be addressed prior its acceptance for publication:

Major comments:

1. In this study, the levels of EO were measured in the cell media. As pointed out by the authors (last paragraph of the discussion) it is not clear what is measured, release? biosynthesis? inhibition of degradation (not mentioned)? This has to be clarified at the beginning of the results section explaining that the “release” is used loosely without real evidence that this is the process that is investigated.

2. The description of ANP receptors and second messengers, which is mentioned in the discussion (second paragraph) should be part of the introduction.

3. The additivity in ANP and ODQ effects do not prove exclude the possibility that ANP acts through activation of GC and that under conditions of complete GC inhibition (in the presence of ODQ) other mechanisms are activated. Examples from the literature showing the use of the inhibitor as evidence for GC participation is appropriate.

4. It is stated that serum levels of ANP is 50 pg/ml and ref 37 is cited (Methods, Treatments). This reference does not state that.

5. The lack of concentration-dependence in both, the effect of ANP on EO levels and its effect on cGMP levels raises doubts as to the validity of the data. How many experiments were done? Were the experiments performed in duplicates, triplicates? What are the error bars in the graphs? SD? SE? Do you know of another example in which cGMP levels showed such a response to a stimulant?

6. Control is missing… for example adding another peptide to the cell’s medium. Also, why EO in the cells was not measured?

7. The authors mentioned the original only study that demonstrated the “release” of EO from H295R cells (ref 12). But, in that study, if I understand correctly, no EO was detected in the conditioned media without dB-cAMP. Was dB-cAMP added in the present study? Is there an agreement as to the quantities of EO “released” from the cells in the two studies?

8. The authors claim that the influence of ANP on EO release was not examined previously – this is not accurate – for example, Crabos et al. have shown that ANP injected intravenously, or included in the in vitro in incubation media of brain slices, decreased the release of EO (Am J Physiol 254:F912, 1988). This has to be mentioned already in the introduction.

Minor comments

1. Abstract, line 13- you mean pg/ml

2. Introduction, line 11. The review of Fedorova et al Biochim Biophys Acta. 1802,2010 is appropriate here

3. Introduction, 3rd paragraph- “striking findings”- why is it so striking?

4. Introduction, 4th paragraph- 54 delete

5. Introduction, 5th paragraph- “EO is secreted when… [33]” it may sound that ref 33 shows this- which of course it does not- correct

6. Methods, Treatments, 1st paragraph- delete-

7. Results, 1st paragraph- “following an early burst”- not clear, where is the burst? It is the first measurement with some basal levels.

8. Results, 2nd paragraph- “For example”- this is not an example, this is the only difference you see.

9. In the three figures – delete “ml” from the y abscise

10. Change the title of all the graph to represent the data shown-not the conclusion from the experiment.

11. Reference 34 was not used in the text.

6. PLOS authors have the option to publish the peer review history of their article (what does this mean?). If published, this will include your full peer review and any attached files.

Reviewer #1: No

Reviewer #2: No

---

## [Author Response · Author response to Decision Letter 0]

23 Oct 2021

3 October 2021

Luis Eduardo M Quintas, Ph.D.

Academic Editor

PLOS ONE

Re: PONE-D-21-16196

Inhibition of Endogenous Ouabain by Atrial Natriuretic Peptide is a Guanylyl Cyclase Independent Effect

Dear Dr. Quintas:

 Thank you for the timely and thoughtful review of our above referenced manuscript. We have taken all of the reviewers’ recommendations in the revision which is being submitted herein. This letter outlines all of the changes made in the manuscript. 

Editor’s Comments

 Response: The corresponding author reviewed the requirements and made all appropriate changes.

'I have read the journal's policy and Dr. El-Mallakh is on the speakers' bureaus of Eisai, Indivior, Intra-Cellular Therapies, Janssen, Lundbeck, Noven, Otsuka, Sunovion, and Teva. The other authors of this manuscript do not have any competing interest.'

Response: The updated statement has been added at the end of the manuscript: 

“Competing Interests

We have read the journal's policy and Dr. El-Mallakh is on the speakers' bureaus of Eisai, Indivior, Intra-Cellular Therapies, Janssen, Lundbeck, Noven, Otsuka, Sunovion, and Teva. The other authors of this manuscript do not have any competing interest.”

Reviewer #1

The authors present an interesting work on endogenous ouabain release in vitro.

The main concern for me is the method used to determine the ouabain concentration. The authors refer to it as radioimmunoassay and cite a paper from 1994. Ouabain is a steroid compound and it is difficult to measure by immune detection.

Let me be very clear; it is not that I don't believe the authors. I do. I also want to say that I think the findings which the authors note are probably "real" and worth communicating. That said, I think the methodology for blood processing and ouabain measurement should be very clearly stated. Specifically, this reviewer would like to see....

1. Information on the antibodies used for the measurements

Response: This is now added to the methods and results sections.

2. Standard curves for ouabain measurement used in assigning values to plasma values.

Response: An example of the standard curve used for these experiments is shown in the new figure 1. All data are expressed as a function of the mass of cells in the flask as reflected in total measured protein. There are no measurements using plasma in this article.

3. Method for anticoagulation of blood samples and details regarding the handling of samples prior to measurement.

Response: None of the samples utilized blood. Further, fetal bovine serum was eliminated from the flasks before each experiment. Any additional information of relevance is now included.

4. Some analysis of elutation fractions from the C18 columns loaded with pooled samples. This should be done with and without spiking with reference ouabain.

Response: This information has been added to the methods section. “In brief, 200 mg Bond Elut C18 columns (Varian) were preconditioned by sequential passage of 3 mls each of 100, 50, 25, and 5% CH3CN in water. Thereafter, each column was washed twice with 3 ml water. For the samples, 10-12 ml of the conditioned cell culture fluid was centrifuged at 2,500 x g for 20 min. The supernatants were applied to the columns which were then washed sequentially with 2 x 3 ml water and 3 ml 5% CH3CN. Differential elution was then used to elute the desired bound steroids as follows: Bound EO was eluted by a 20% CH3CN wash, while less polar steroids including endogenous digoxin-like immunocrossreactive materials [9,51] and aldosterone, that remained bound after the 20% CH3CN wash were eluted with a 50% CH3CN wash. Sample blanks were run in parallel. In some experiments, the recovery of EO as well as the success of the differential elution was assessed by inclusion of tracer amounts of 3H-ouabain and 3H-digoxin. For immunoassay, all eluates of interest were dried by vacuum centrifugation (Savant) and reconstituted at 60-120 x their original concentration in water. For analysis of the efficiency of steroid recoveries, aliquots of the eluates were taken directly for liquid scintillation counting. The radioimmunoassay was performed in a manner similar to that previously described using the R8 ouabain-antiserum (38) with the exception that a micro format was employed wherein the total assay volume was 50 µL. In this assay, 30 µL was the reconstituted sample and the remainder was comprised of 3H-ouabain (final concentration = 1.35 nM) and buffer. The reaction was incubated at room temperature for one hour and terminated by rapid filtration over glass fiber filters (Whatman GF/B) using a 24 well cell harvester (Brandel). The filters were soaked in scintillation cocktail (Ultima Gold, Perkin Elmer) for 40 hours and then counted (Beckman TA3000) with quench correction. Non-specific binding was determined by inclusion of excess ouabain in the reaction and typically was < 4 % of the total bound count. The antiserum exhibits no meaningful crossreactivity (<0.01%) with the vast majority of common adrenocortical, ovarian or testicular steroids (see Table 1 in [51]). In addition, the immuno-crossreactivities for dihydro-ouabain, digoxin, digitoxin, and aldosterone were 0.16, 5.2, 28 and 0.012 %, respectively. The crossreactivity of the latter three steroids is of no concern; the use of a differential elution process excludes the vast majority of digoxin, digoxin-like materials, and aldosterone from the eluates used for EO measurement. The low crossreactivity for dihydro-ouabain, i.e., 625-fold less than for ouabain, makes it unlikely that any dihydro-ouabain that may have been secreted [11,12] contributed significantly to the EO measurement.”

5. Greater discussion of the challenges inherent in the immunoassay of ouabain.

Response: An extensive discussion of most of the critical issues has been added to the text. “The existence of endogenous cardenolides, like EO, remains controversial to some [55] despite overwhelming evidence of its existence and physiological and pathological roles - some of which has been proven in studies using transgenic animals [21]. Many of the challenges related to the successful measurement of EO have been described elsewhere [21]. Of critical importance is the extraction procedure which must be performed with care. Ouabain is a highly polar steroid; its interaction with C18 is much weaker than all of the known adrenocortical steroids and most of the known CTS. Accordingly, it is critical to fully precondition the C18 columns, bringing them carefully and fully to water before sample application to ensure that all of the sample EO will become bound. The presence of even small amounts of organic solvents such as acetonitrile or methanol will prevent EO from binding and it will be lost to the flow through. Further, the mass of the C18 sorbent must be sufficient to trap all the EO and this is a function of the sample mass, type, and origin. For example, 200 mg of C18 sorbent will typically trap all the EO from 2-3 ml of plasma. With conditioned cell culture medium, up to 20 ml can be extracted per column because this matrix is much less complex than native plasma and the competition of ancillary polar chemical entities with EO for binding to the sorbent is much less. Once the sample has been applied and EO is bound, it is necessary to wash with the appropriate volumes of water and then low concentrations of acetonitrile (2.5 to 5%) or methanol (5-10%) to reduce some of the very highly polar entities that can interfere in the radioimmunoassay and lead to massive suppression of ionization in mass spectrometry. The volume of this organic wash also is critical; overly large wash volumes will eventually elute bound EO and it will be lost. Similarly, the concentration and volume of acetonitrile used to elute EO is important. Typically, 20-25% acetonitrile is used and the volume is chosen to ensure that all the EO is recovered without significant contamination by e.g., digoxin (Figure 1) or other less polar eCTS. In validating the extraction method, it is most useful to use 3H labeled steroids of interest using the anticipated sample conditions. For the immunoassay, a highly selective antiserum with affinity for ouabain in the high picomolar to low nanomolar range is required that has minimal crossreactivity with the common classical mammalian steroids. Thus far, this antibody requirement has been met with polyclonal but not monoclonal sources. The final critical factor in the radioimmunoassay method we use concerns the method by which the binding assay is terminated. With the vast majority of high affinity ouabain antibodies, the half time for the dissociation of ouabain or EO from antibody binding sites is surprisingly rapid ranging from 2-15 minutes. Thus, the method used to separate bound from free must be much faster than the dissociation half time if the high sensitivity required to detect EO is to be retained. Among the various options, filtration over glass fiber filters is convenient and very rapid, typically requiring <15 seconds including multiple washes. Another advantage of the method is the very low background binding of the tracer 3H-ouabain to the filters. Other assay formats including ELISA [51,56], radioimmunoassays that do not require a bound/free separation as well as mass spectrometry [21] can be used with the appropriate modifications.”

6. The authors describe that the cells were treated with 0.1% DMSO or ANP. Was the ANP given also in DMSO solution? This sentence suggest otherwise.

Response: The sentence was clarified as follows: “H295R human adrenocortical cells were treated with: 1) 0.1% DMSO or ANP (5, 50, 200 pg/ml; California Peptide Research, Inc, Salt Lake City, UT) dissolved in 0.1% DMSO for 30 min, 60 min and 90 min; . . . ” 

Reviewer #2 

This study by Tegin et al. describes the effect of ANP on the levels of endogenous ouabain in the media of H295R cells. The possible involvement of Guanylyl cyclase in this effect was addressed by studying the effect of ANP following pretreatment with Guanylyl cyclase inhibitor ODQ.

Although the paper describes interesting observations several key issues have to be addressed prior its acceptance for publication:

Major comments:

1. In this study, the levels of EO were measured in the cell media. As pointed out by the authors (last paragraph of the discussion) it is not clear what is measured, release? biosynthesis? inhibition of degradation (not mentioned)? This has to be clarified at the beginning of the results section explaining that the “release” is used loosely without real evidence that this is the process that is investigated.

 Response: We appreciate the reviewer pointing out this oversight on our part. We added a paragraph at the end of the Results section to address this: “While we use the term ‘secretion,’ our data does not determine the source of the EO, other than the H295R cells themselves, that was measured. For example, we do not know if the spontaneous appearance of EO in the medium conditioned by the cells is derived from de novo synthesis, release of stored EO, or even if there is a degradation of a carrier protein that obscures EO measurements at baseline. It may be noted that, in the adrenal literature in general, it is almost an invariable and reasonable assumption that the appearance of a steroid in increasing concentrations in the cell culture medium is a reflection of biosynthesis followed by net secretion. In this article, the suppression of EO secretion by ANP may be related to any of the abovementioned phenomena..” However, it may be noted that the ability of ANP to suppress aldosterone secretion is known to be a reflection of diminished biosynthesis. We feel it is most likely that ANP suppresses EO section by the same general phenomenon. 

2. The description of ANP receptors and second messengers, which is mentioned in the discussion (second paragraph) should be part of the introduction.

 Response: The entire second paragraph was moved to the Introduction.

3. The additivity in ANP and ODQ effects do not prove exclude the possibility that ANP acts through activation of GC and that under conditions of complete GC inhibition (in the presence of ODQ) other mechanisms are activated. Examples from the literature showing the use of the inhibitor as evidence for GC participation is appropriate.

 Response: We appreciate the reviewer’s comments here and they impact the discussion of point #5 as well. We added the following sentences at the end of the first paragraph of the Discussion: “Nonetheless, there are potential alternative explanations of the data. For example, nitric oxide (NO) is known to activate guanylyl cyclase and sufficient NO can overcome ODQ inhibition [52]. Likewise, confirmation of a guanylyl cyclase component does not exclude other potential mechanisms of ANP, as is the case with NO [43,53]. This is especially important given the lack of a clear dose response effect of ANP on either EO or cGMP production.”

4. It is stated that serum levels of ANP is 50 pg/ml and ref 37 is cited (Methods, Treatments). This reference does not state that.

 Response: The paper reports that the normal range of human ANP levels is “75 to 225 pg/ml.” Most reports of ANP levels in normal humans are in the mid 30s with pathological levels having a larger range up to the low 200s. We added multiple primary references and changed the statement as follows: “Normal serum ANP levels are frequently reported in the mid 30s pg/mL with a range up to 200 pg/mL in renal failure [47-49].”

5. The lack of concentration-dependence in both, the effect of ANP on EO levels and its effect on cGMP levels raises doubts as to the validity of the data. How many experiments were done? Were the experiments performed in duplicates, triplicates? What are the error bars in the graphs? SD? SE? Do you know of another example in which cGMP levels showed such a response to a stimulant?

 Response: All experiments were done in triplicates to sextuplicates. This was indicated in the ‘Treatments’ section of Methods: “All experiments had 3 – 6 replicates.” And The data were presented as mean ± SE. We added this in the Statistics section: “The data were presented as mean ± SE”

 Also all the EO measurements were made blinded. This was also added in the Methods section: “All measurements were made with an operator blinded to the study design.”

There are previous studies that suggest that there is a rapid acclimation (tachyphylactic) to ANP effect in physiologic systems. The transient natures of our dose response curves could reflect that tachyphylaxis. We addressed this at the end of the 2nd paragraph of the discussion: “Conversely, the transient nature of effects of ANP on both EO and cGMP may be consistent with the apparent tachyphylaxis seen in blood pressure control with ANP [54].”

6. Control is missing… for example adding another peptide to the cell’s medium. Also, why EO in the cells was not measured?

 Response: We did not have a random peptide control. We had zero treatment controls, i.e., baseline only. We did not measure EO inside cells because intracellular concentrations were beyond the sensitivity of our previous assays. We indicated the former in the ‘Treatments’ section of the ‘Methods’: “We only used no treatment controls, and did not examine the effect of adding other non-ANP peptides to the cell cultures.” 

 We indicated the latter in the ‘Measurements’ section of the ‘Methods’: “We did not assay intracellular EO concentrations because previous work revealed that the intracellular EO concentrations in H295R cells was below assay detection [12].”

It is certainly possible to measure EO in H295R cells but the number of cells required for reliable measurements would be at least 10-fold greater than we used here.

7. The authors mentioned the original only study that demonstrated the “release” of EO from H295R cells (ref 12). But, in that study, if I understand correctly, no EO was detected in the conditioned media without dB-cAMP. Was dB-cAMP added in the present study? Is there an agreement as to the quantities of EO “released” from the cells in the two studies?

 Response: (Bu)2 cAMP is only needed for dihydro-ouabain, EO is secreted without its addition. No changes were made. 

We replicated the findings of the previous study but our treatment times were much briefer and we expressed our findings in different units (pg/mL/protein instead of pg/mL/106 cells). No changes were made to the text since we had already mentioned that we replicated the previous study in the first paragraph of the Discussion.

8. The authors claim that the influence of ANP on EO release was not examined previously – this is not accurate – for example, Crabos et al. have shown that ANP injected intravenously, or included in the in vitro in incubation media of brain slices, decreased the release of EO (Am J Physiol 254:F912, 1988). This has to be mentioned already in the introduction.

Response: We very much appreciate the reviewer pointing this out. We added this reference in the Introduction: “Over 30 years ago, ANP was demonstrated to reduce synthesis of a Na+-K+-ATPase inhibitor by brain tissue both in vivo and in vitro [45].”

Minor comments

1. Abstract, line 13- you mean pg/ml

 Response: Yes, thank you. That has been corrected. 

2. Introduction, line 11. The review of Fedorova et al Biochim Biophys Acta. 1802,2010 is appropriate here

 Response: This reference was added.

3. Introduction, 3rd paragraph- “striking findings”- why is it so striking?

 Response: The adjective was removed so the sentence now reads: “Both ANP and ECs are thought to play important roles in sodium and fluid homeostasis.”

4. Introduction, 4th paragraph- 54 delete

 Response: done

5. Introduction, 5th paragraph- “EO is secreted when… [33]” it may sound that ref 33 shows this- which of course it does not- correct

 Response: The reviewer is correct. We deleted the sentence and the reference.

6. Methods, Treatments, 1st paragraph- delete-

 Response: We are unclear what the reviewer would like us to delete. We do not think that deleting the entire paragraph is what the reviewer had in mind.

7. Results, 1st paragraph- “following an early burst”- not clear, where is the burst? It is the first measurement with some basal levels.

Response: That appears to have been an oversight. Thank you. The sentence has been corrected: “The concentration in the supernatant spontaneously increases over time, and nearly doubles in 90 minutes (Figure 1).”

8. Results, 2nd paragraph- “For example”- this is not an example, this is the only difference you see.

 Response: “For example, . . .” was replaced with “Specifically, . . .” 

9. In the three figures – delete “ml” from the y abscise

 Response: We added the statement “Y axis units are pM/mg protein.” 

10. Change the title of all the graph to represent the data shown-not the conclusion from the experiment.

 Response: The Figure legends were all changed as recommended, as follows:

(Note: the original figure numbers shown below have all incremented in the revised version due to the addition of a new figure 1). 

“Figure 1. The EO content in the culture media of H295R human adrenocortical cells increased linearly over the time indicated significantly at 90 min. in the absence of ANP treatment. ANP concentrations of 5, 50, 200 pg/mL reduced EO secretion at 60 and 90 min. The inhibitory effect of 50 pg/mL ANP was significantly lower compared to lower (ANP5) and higher (ANP200) at 90 min. Error bars represent standard deviation. (*P < 0.05, compared to 30 min untreated, **P < 0.05, compared to untreated at corresponding time point, ***P < 0.05, compared to ANP5 and ANP200 at 90 min. Y axis units are pM/mg protein.)”

“Figure 2. Intracellular cGMP levels following ANP treatment for 30, 60, and 90 min in H295R human adrenocortical cells. A significant increase in cGMP was observed only with the physiologic concentration of 50 pg/mL at 60min. Error bars represent standard deviations. (*P < 0.05 compared to control. Y axis units are pM/mg protein.)”

“Figure 3. Change in EO secretion 60 min following addition of ANP 50 pg/mL, 10 mM ODQ and ODQ+ANP treatment. ANP alone and ODQ alone reduced EO secretion (#P < 0.05 compared to control). Pretreatment with ODQ followed by ANP further decreased secreted EO when compared to ANP or ODQ alone and their inhibitory effects appeared to be additive (*P < 0.05; Y axis units are pM/mg protein.).”

11. Reference 34 was not used in the text.

 Response: The numbering of reference 34 has changed to 43 in this revision, and it has been used in the response to point #3 above. 

12. Data is now available on Zenodo a under the title and authors of this publication (and may be accessed at: https://zenodo.org/record/5594568#.YXRY-S2z3a8).

We would greatly appreciate the re-review of our submission for publication in PLoS One.

Thank you,

Sincerely,

Rif S. El-Mallakh, MD

(corresponding author)

---

## [Decision Letter · Decision Letter 1]

4 Nov 2021

Inhibition of Endogenous Ouabain by Atrial Natriuretic Peptide is a Guanylyl Cyclase Independent Effect

PONE-D-21-16196R1

Dear Dr. El-Mallakh,

We’re pleased to inform you that your manuscript has been judged scientifically suitable for publication and will be formally accepted for publication once it meets all outstanding technical requirements.

Kind regards,

Luis Eduardo M Quintas, Ph.D.

Academic Editor

PLOS ONE

Additional Editor Comments (optional):

Reviewers' comments:

Reviewer's Responses to Questions

**Comments to the Author**

1. If the authors have adequately addressed your comments raised in a previous round of review and you feel that this manuscript is now acceptable for publication, you may indicate that here to bypass the “Comments to the Author” section, enter your conflict of interest statement in the “Confidential to Editor” section, and submit your "Accept" recommendation.

Reviewer #1: All comments have been addressed

Reviewer #2: All comments have been addressed

2. Is the manuscript technically sound, and do the data support the conclusions?

Reviewer #1: Yes

Reviewer #2: Yes

3. Has the statistical analysis been performed appropriately and rigorously? 

Reviewer #1: Yes

Reviewer #2: Yes

4. Have the authors made all data underlying the findings in their manuscript fully available?

Reviewer #1: Yes

Reviewer #2: Yes

5. Is the manuscript presented in an intelligible fashion and written in standard English?

Reviewer #1: Yes

Reviewer #2: Yes

6. Review Comments to the Author

Reviewer #1: (No Response)

Reviewer #2: (No Response)

7. PLOS authors have the option to publish the peer review history of their article (what does this mean?). If published, this will include your full peer review and any attached files.

Reviewer #1: No

Reviewer #2: No

---

## [Editor Report · Acceptance letter]

8 Nov 2021

PONE-D-21-16196R1 

Inhibition of Endogenous Ouabain by Atrial Natriuretic Peptide is a Guanylyl Cyclase Independent Effect 

Dear Dr. El-Mallakh:

I'm pleased to inform you that your manuscript has been deemed suitable for publication in PLOS ONE. Congratulations! Your manuscript is now with our production department. 

Kind regards, 

on behalf of

Dr. Luis Eduardo M Quintas 

Academic Editor

PLOS ONE